# Neural USD: An object-centric framework for iterative editing and control

## Abstract

Amazing progress has been made in controllable generative modeling, especially over the last few years. However, some challenges remain. One of them is precise and iterative object editing. In many of the current methods, trying to edit the generated image (for example, changing the color of a particular object in the scene or changing the background while keeping other elements unchanged) by changing the conditioning signals often leads to unintended global changes in the scene. In this work, we take the first steps to address the above challenges.

Taking inspiration from the Universal Scene Descriptor (USD) standard developed in the computer graphics community, we introduce the "Neural Universal Scene Descriptor" or Neural USD. In this framework, we represent scenes and objects in a structured, hierarchical manner. This accommodates diverse signals, minimizes model-specific constraints, and enables per-object control over appearance, geometry, and pose. We further apply a fine-tuning approach which ensures that the above control signals are disentangled from one another. We evaluate several design considerations for our framework, demonstrating how Neural USD enables iterative and incremental workflows.

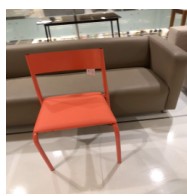 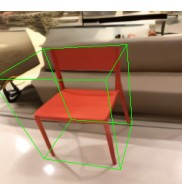 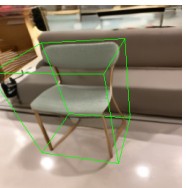 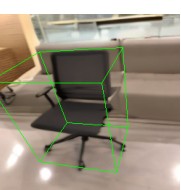 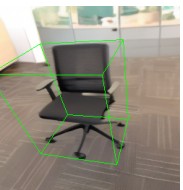

(a) Original    (b) Pose (cam & object)    (c) Appearance    (d) Geometry    (e) Background

Figure 1: Demo of iterative editing using the finetuned Neural USD model. Given the original image (a), we specify the camera pose and desired target obj pose as a 3D bounding box (b). The model is able to precisely 'move' the object to the desired pose while rest of the scene elements remain fairly consistent. Next in (c), we change the appearance (in this case color) while also being in the new pose. Our model is able to handle these multiple requests. In (d), we retain the desired pose from (b) and condition on another office chair's geometry (or depth) and appearance. Our model is able to perform these edits while leaving rest of the scene elements (notably the background) consistent with the original image. Note that using these conditions we can replace an object in the image with another object in any desired pose. In (e), we edit the pose, geometry and background all at the same time. Our model is able to handle these requests simultaneously and generates an image that respects all the given conditions. More details on the conditioning signals and additional examples are in Section 4.2 and appendix E

.

## 1 Introduction

The surge in relevance of visual generative models has led to the development of a wide range of conditioning approaches. These approaches enable control over generated outputs by allowing users to guide the generation process using textual prompts, reference images, or other forms of input like a desired depth map, segmentation mask, edge map and others. However, conditioning choices are

often tailor-made for specific model architectures, or limit the user to global scene edits, restricting portability across models and limiting users from performing object-level, incremental updates to their content.

Several approaches have been proposed to address the challenge of conditioning and controlling visual generative models. One area of study investigates how models can be conditioned on depth maps, edge maps, segmentation maps, and other conditioning signals (Zhang et al., 2023; Mou et al., 2023; Avrahami et al., 2022; Li et al., 2023b). These approaches allow the user to control one aspect of the scene at a time. However, these methods can result in global scene changes as a result of local conditioning signal edits. This problem compounds as we try to make multiple edits to the scene. In contrast, our framework enables multiple serial edits to the original image while preserving other visual elements not part of the edit. Another line of work uses text prompts to guide image generation and editing (Brooks et al., 2022; Rombach et al., 2022). While these approaches generate impressive results, they often limit what a user is able to express due to the challenge of describing complex scene layouts with text. Recent approaches propose object-centric conditioning formats to guide image generation (Bhat et al., 2023b; Wu et al., 2024; Liu et al., 2023a; Michel et al., 2023). These methods often struggle with handling multiple objects in a scene, or are limited in supporting conditioning modalities beyond reference images.

To address these challenges, we introduce Neural Universal Scene Descriptors (Neural USD): an object-centric conditioning standard that enables precise control of geometry, appearance, and object poses within generative models. The Neural USD is defined as an XML-style format consisting of per-object attributes (see Fig. 3 (a)). These include (1) *appearance* - encoded as feature vectors from some pre-trained embedding methods like CLIP or Dino (2) *position* - encoded as a 2d or 3d bounding box around the desired object. (3) *geometry* - includes features like depth map, segmentation mask, and edge map. In the experiments in this paper, we pre-dominantly used depth-map for gemoetric features. Each of these modalities is tokenized into a sequence of conditioning vectors and passed to downstream generative models for conditioning. This representation is compatible with many architectures including diffusion models (Sohl-Dickstein et al., 2015; Ho et al., 2020), DiTs (Peebles & Xie, 2023), and transformers (Vaswani et al., 2017; Yu et al., 2022) - thereby facilitating cross-model portability. After fine-tuning the image models with the Neural USD data, we can manually edit the objects and backgrounds in the image using a simple recipe: $\mathcal{I} = \text{Decode}(\text{Edit}(\text{USD}))$. However, naïve training on such a representa-

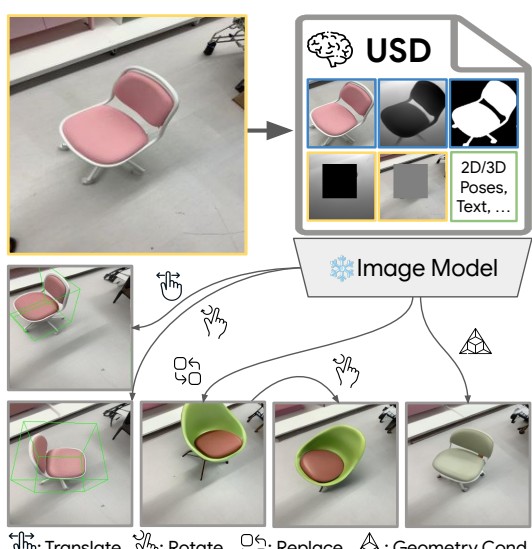

: Translate   : Rotate   : Replace   : Geometry Cond.

Figure 2: Neural USD enables computer graphics-style control of image models. A Neural USD represents an image as assets with appearance, geometry, and pose. Fine-tuning adapts pre-trained models to these signals while keeping appearance and geometry pose-invariant.

tion would cause challenges, as the model would have difficulty disentangling pose and appearance attributes of the conditioning signal, empirically resulting in poor object control. We solve this by training the model using a pair of images (say source, target) from a video sequence. In this case, the goal of the model is to generate the target image; the geometric and appearance conditioning comes from the source image and the pose conditioning from the target image. This results in robust object pose, geometry, and appearance control signals that are dis-entangled from one another.

In summary, our main contributions are as follows:

1. Neural USD (Fig 2): an object-centric conditioning format for a broad class of generative models that provides precise control over object position, appearance, and geometry (Fig 3 (a)). We also show how to finetune existing models using pairs of images from videos

to enable learning disentangled control signals across all modalities. Our framework is generic and applicable to a wide class of generative models.

2. Our method enables iterative editing workflows where the target object changes in accordance with the conditioning signal and other scene elements remain fairly unchanged and/or consistent with the original image (Figs. 1 and 4).

3. We also compare our method against several standard baselines. Our experiments show that Neural USD allows for more precise object control as measured by the reconstruction error (Fig. 8).

The rest of the paper is organized as follows. In Section 2, we discuss other related works and how our approach sits within the broader generative landscape. In Section 3, we describe the Neural USD framework. This includes details on how the different aspects like pose, appearance and geometry are encoded and then combined into a single conditioning signal. We also discuss our approach to fine-tuning and learning from pairs of images. We describe our experimental results in Section 4. In Section 4.1, we give examples of how Neural USD enables object centric editing. In Section 4.2, we give more examples of iterative editing. In Section 4.3, we compare our approach to other generative models. We conclude the paper in Section 5.

## 2 RELATED WORK

Recent work in object-centric learning decomposes visual scenes into distinct object representations for structured, interpretable image generation. Slot-based methods such as Slot Attention (Locatello et al., 2020), SLATE (Singh et al., 2021), and STEVE (Singh et al., 2022b) model scenes as independent entities, with refinements like LSD (Jiang et al., 2023) and Slot Diffusion (Wu et al., 2023) improving disentanglement. These representations support tasks including attribute manipulation (Singh et al., 2022a), motion modeling (Seitzer et al., 2023), and 3D pose estimation (Jabri et al., 2023); OSRT (Sajjadi et al., 2022) further addresses global camera pose. Yet such models struggle with real-world data. Our approach leverages self-supervised visual encoders with large-scale pre-trained diffusion models, enabling scalable object-centric learning in realistic settings.

Personalized image generation has progressed from test-time fine-tuning (DreamBooth (Ruiz et al., 2022), Textual Inversion (Gal et al., 2022)) to zero-shot personalization (InstantBooth (Shi et al., 2023a), ZeroShotBooth (Jia et al., 2023), BLIP-Diffusion (Li et al., 2023a), ELITE (Wei et al., 2023), InstantID (Wang et al., 2024b), FastComposer (Xiao et al., 2023)). While effective, most produce single-subject images without spatial control. Exceptions such as VisualComposer (Parmar et al., 2025), TokenVerse (Garibi et al., 2025), and Video Alchemist (Chen et al., 2025) allow multi-entity composition, but with limited control. Subject-Diffusion (Ma et al., 2023) introduces 2D bounding-box conditioning but lacks 3D pose control. We extend controllability by incorporating object poses, enabling structured multi-object generation and manipulation.

Spatial control in diffusion models is pursued through bounding boxes or segmentation masks. Strategies include prompt manipulation (Kawar et al., 2022; Ge et al., 2023; Brooks et al., 2022), attention adjustments (Xie et al., 2023; Kim et al., 2023; Chen et al., 2023; Chefer et al., 2023; Feng et al., 2022; Hertz et al., 2022; Cao et al., 2023), and latent editing (Epstein et al., 2023; Shi et al., 2023c; Luo et al., 2023). Fine-tuned approaches add spatial conditioning (Gafni et al., 2022; Avrahami et al., 2022; Yang et al., 2022; Hu et al., 2023; Xu et al., 2023; Goel et al., 2023). GLI-GEN (Li et al., 2023b) introduces attention layers for box conditioning, InstanceDiffusion (Wang et al., 2024c) supports masks and scribbles, and ControlNet (Zhang et al., 2023) incorporates depth and normals; Boximator (Wang et al., 2024a) extends these ideas to video.

3D-aware image generation pursues structured scene synthesis. GAN-based methods use explicit 3D representations such as radiance fields (Chan et al., 2020; Gu et al., 2021; Chan et al., 2021; Schwarz et al., 2020; Niemeyer & Geiger, 2020; Xu et al., 2022) and meshes (Chen et al., 2019; 2021; Gao et al., 2022; Pavllo et al., 2020; 2021). Diffusion-based approaches (Shi et al., 2023b; Liu et al., 2023b; Poole et al., 2022; Wang et al., 2022; Lin et al., 2022; Kant et al., 2024; Melas-Kyriazi et al., 2023; Watson et al., 2022) transfer 2D knowledge into 3D. 3DiM (Watson et al., 2022) and Zero-1-to-3 (Liu et al., 2023a) leverage multiview training but remain object-centric. More recent methods address multi-object real-world scenes (Sargent et al., 2023; Pandey et al., 2023; Yenphraphai et al., 2024; Alzayer et al., 2024); OBJect-3DIT (Michel et al., 2023) enables language-guided editing but

is synthetic-data limited. LooseControl (Bhat et al., 2023b) uses 3D bounding boxes as depth maps for pose control, but is not directly applicable to editing.

We unify these directions by enabling both 2D and 3D spatial conditioning in pre-trained diffusion models, with support for appearance and geometry inputs. Using bounding boxes as control signals, our approach enables fine-grained object pose manipulation—including rotation and occlusion—while scaling to multiple modalities and offering a general recipe for incorporating new ones.

# 3 METHOD

Figure 3: Neural USD Overview. a) A Neural USD consists of assets with multiple modalities: appearance, geometry, and pose. b) Pre-trained image models fine-tune on Neural USD, encoding appearance and geometry from a source image and pose from a target image to reconstruct the target. c) At inference, objects' poses, geometry, and appearance can be modified, including the background.

We propose Neural USD as an object-centric representation of a scene, composed of appearance, geometry, and pose representations. Image models are trained to reconstruct target objects defined in the Neural USD by using paired images extracted from video sequences. We additionally apply modality dropout. This allows the model to learn disentangled appearance, geometry, and pose representations. The resulting model allows for fine-grained control of objects in the scene. Such a conditioning format draws parallels to the intuitive object-centric workflows used in computer graphics programs such as Blender (Blender Online Community, 2018).

## 3.1 DATA

In this work, we explore datasets with 2D and 3D annotations readily available. Obtaining tracked 2D bounding boxes for video is a problem with promising solutions (Li et al., 2024). However, obtaining 3D bounding box annotations at scale is still an open challenge, but may be addressed in the near future given improvements in SLAM, point tracking, and depth estimation (Zhang et al., 2024; Bhat et al., 2023a; Yang et al., 2024; Doersch et al., 2023).

We compose the Neural USD dataset by applying separate annotation models to the original datasets. We acquire depth annotations by applying ZoeDepth (Bhat et al., 2023a) then crop and normalize per-object depth maps. Object masks are computed by applying SAM (Kirillov et al., 2023) with bounding box conditioning. Additional conditioning signals such as surface normals and point-clouds can be extracted with open source models (Yang et al., 2023), though we leave this for future work.

## 3.2 ASSETS IN NEURAL USD

We borrow the nomenclature of Neural Assets (Wu et al., 2024) to describe the components of the Neural USD. A Neural USD is defined as a list of $N$ assets $\{\hat{a}_1, ..., \hat{a}_N\}$, where each asset $\hat{a}_i$ is defined as a tuple of attributes such as 2D or 3D bounding box coordinates $\mathcal{P}_i^{2D}$, $\mathcal{P}_i^{3D}$, appearance

descriptors such as image crops or clip embeddings $\mathcal{A}_i$, geometry signals from depth images, masks, pointclouds, and surface normals $\mathcal{G}_i$, or even text $\mathcal{T}_i$. The resulting asset can be defined as the tuple $\hat{a}_i = (\mathcal{P}_i^{\text{2D}}, \mathcal{P}_i^{\text{3D}}, \mathcal{A}_i, \mathcal{G}_i, \mathcal{T}_i, ...)$.

## 3.3 ENCODING ASSETS

To make the Neural USD a compatible conditioning format for arbitrary downstream models, it must first be encoded into a continuous vector representation such that the encoded appearance and geometry descriptors can be defined as continuous vectors $\mathcal{A}_i^{\text{emb}}, \mathcal{G}_i^{\text{emb}} \in \mathbb{R}^{\text{K} \times \text{D}}$, and pose embeddings as $\mathcal{P}_i^{\text{emb, 2D}}, \mathcal{P}_i^{\text{emb, 3D}} \in \mathbb{R}^{\text{D'}}$. This token representation enables the fine-tuning of arbitrary model architectures with the Neural USD encoding, as well as separate control of pose, geometry, and appearance. We now describe how appearance, geometry, and poses are encoded in this format.

### 3.3.1 APPEARANCE AND GEOMETRY ENCODING

Obtaining $\mathbb{R}^{\text{K} \times \text{D}}$ encodings of appearance and geometry can vary depending on the modality being handled. Often times modalities can have varying dimensions (images vs. pointclouds) or different semantic meaning (surface normals vs. depth). As such, we utilize separate encoders for each modality in the Neural USD. In the case of appearance signals in the form of images $\mathcal{I} \in \text{H} \times \text{W} \times \text{C}$ we apply a pre-trained DINOv2 (Caron et al., 2021) model to obtain output features $\mathcal{F} = \text{Encoder}(\mathcal{I}_i)$, where $\mathcal{F} \in \text{h} \times \text{w} \times \text{D}$. The first two dimensions are then flattened to obtain the resulting embedding $\mathcal{A}_i^{\text{emb}} \in \mathbb{R}^{\text{K} \times \text{D}}$. Similarly, geometry features such as surface normals and depth can be processed using a separate pretrained DINOv2 backbone. We find that normalizing depth features on a per-object basis leads to improved generalization performance, as raw metric depth signals constrain the object to certain locations in the scene. Preliminary experiments using a shared backbone for both image and depth yielded suboptimal results, as the model struggled to accurately represent both geometry and appearance.

Approaches such as Neural Assets (Wu et al., 2024) first embed an image with a pre-trained backbone and slice the resulting feature map using the corresponding 2D bounding box locations for each object. While this results in fewer forward passes, it leads to challenges when replacing objects in the scene, or conditioning on modalities such as depth or points, since object features are globally correlated. Instead, we process each object appearance and depth feature separately, removing global correlations. This can be done efficiently by pre-computing these features and storing them in the Neural USD.

### 3.3.2 2D AND 3D POSE ENCODING

Utilizing a separate encoding for 2D and 3D pose signals provides the user access to two different interfaces for controlling the position of the object in the scene. The 2D pose allows for simple dragging of the object around the scene, while the 3D pose allows for more sophisticated control of properties such as distance from the camera and rotation. The 2D bounding box is defined as the image-normalized coordinates of the top left corner of the bounding box, as well as the image-normalized height and width $p_i^{\text{2D}} = (x_i, y_i, h_i, w_i)$. We represent the 3D bounding box by projecting four corners that span the bounding box to the image plane, arriving at $\{p_i^{\text{3D},j} = (h_i^j, w_i^j, d_i^j)\}_{j=1}^4$, with projected image-normalized 2D coordinates $(h_i^j, w_i^j)$ and 3D depth $d_i^j$. We project the 2D bounding box and the concatenated corners of the 3D bounding box with a simple multi-layer perceptron to obtain:

$$\mathcal{P}_i^{\text{emb, 2D}} = \text{MLP}(p_i^{\text{2D}}), \tag{1}$$

$$\mathcal{P}_i^{\text{emb, 3D}} = \text{MLP}(p_i^{\text{3D}}), \tag{2}$$

$$p_i^{\text{3D}} = \text{Concat}[p_i^{\text{3D},1}, p_i^{\text{3D},2}, p_i^{\text{3D},3}, p_i^{\text{3D},4}]. \tag{3}$$

## 3.4 COMBINING ENCODINGS

In this section, we describe how we combine the defined encodings so that they can be used to condition downstream models via cross-attention (Vaswani et al., 2017), FiLM layers (Perez et al., 2017), or in place of text embeddings. To do so, we simply concatenate the appearance, geometry,

and pose tokens channel-wise.

$$\tilde{a}_i = \text{Concat}[\mathcal{A}_i^{\text{emb}}, \mathcal{G}_i^{\text{emb}}, \mathcal{P}_i^{\text{emb, 3D}}, \mathcal{P}_i^{\text{emb, 2D}}], \tag{4}$$

$$\tilde{a}_i \in \mathbb{R}^{\text{K} \times \text{M}}. \tag{5}$$

where the Neural USD asset encoding $\tilde{a}_i$ is projected via an MLP to obtain:

$$a_i = \text{MLP}(\tilde{a}_i), a_i \in \mathbb{R}^{\text{K} \times D}. \tag{6}$$

Finally all Neural USD asset encodings are concatenated along the token dimension to obtain the final Neural USD encoding:

$$\mathcal{N} = \text{Concat}[\tilde{a}_i, ..., \tilde{a}_N], \mathcal{N} \in \mathbb{R}^{(\text{N} \times K) \times D}. \tag{7}$$

Although approaches such as Neural Assets (Wu et al., 2024) require the background to be encoded separately, the flexible structure of the Neural USD allows the background to simply be defined as another asset, with its corresponding appearance and geometry signals. Masking foreground objects in the provided background signals led to improved results. We provide an additional pose embedding during training to represent the source-to-target transform. This helps the model learn not only the movement of the objects, but also that of the background scene (i.e. camera movement).

## 3.5 FINE-TUNING MODELS WITH NEURAL USD

Neural USD makes few assumptions about the downstream image model to be fine-tuned with the Neural USD. Given that a Neural USD is simply a sequence of tokens, it is amenable to conditioning via cross-attention or FiLM layers; techniques supported by nearly all architectures currently used in generative modeling. In this work, we use Stable Diffusion v2.1 (Rombach et al., 2022), an exemplary open source generative model which is widely used. Given the relatively poor performance of Stable Diffusion v2.1 compared to SOTA models, we do not expect SOTA-level prediction quality and instead demonstrate new conditioning capabilities that can be applied to image models. Both the encoders and the image model are fine-tuned end to end using the training objective defined in the following section. During training, we randomly zero out tokens for the entire asset, for modalities of an asset, or for 2D and 3D pose signals. This helps individual modality features to be invariant of other modality features, allowing for precise control. Modality dropout also allows the use of Classifier Free Guidance (Ho & Salimans, 2022) during evaluation.

## 3.6 LEARNING FROM PAIRS OF IMAGES

The naïve approach of using individual images with Neural USD annotations leads to the entanglement of conditioning signals, whereby the model only uses the appearance and geometry encodings and entirely disregards the pose encodings, thereby limiting pose control of objects in the scene. The use of video sequences yields a promising solution to this challenge. Video sequences offer multiple views of objects in the scene, granting us access to a variety of sources information when constructing our Neural USD encoding. Specifically, we extract appearance, geometry, and other spatial conditioning signals from a source image $I_{\text{src}}$ by cropping out these elements with the corresponding 2D bounding-box annotations. The 2D and 3D target poses are referenced from a target image $I_{\text{tgt}}$. The Neural USD encoding is composed using the source spatial modalities and target poses and provided to the image model. The training objective is to reconstruct $I_{\text{tgt}}$ using the denoising diffusion loss of Stable Diffusion v2.1 (Rombach et al., 2022). This training recipe encourages the model to learn appearance and geometry encodings that are not correlated with pose encodings. The resulting model can be controlled via multiple independent control signals.

## 4 EXPERIMENTS

Through our experiments, we try to answer the following questions: 1) Does Neural USD allow for precise object pose, appearance and geometry control? 2) Does Neural USD allow multiple edits to be made to an object while keeping other elements unchanged? 3) How does Neural USD compare to other generative model conditioning approaches?

For all our experiments, we used the following datasets.

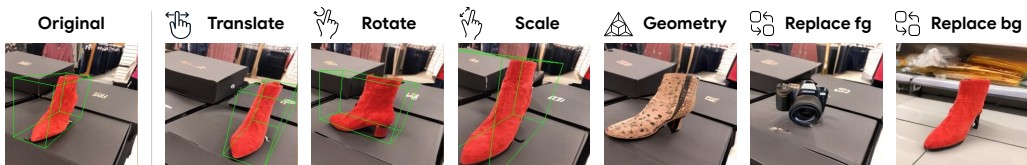

Figure 4: Neural USD allows users to perform a variety of pose, appearance, and geometry modifications to both the foreground and the background objects. Camera pose remains fixed.

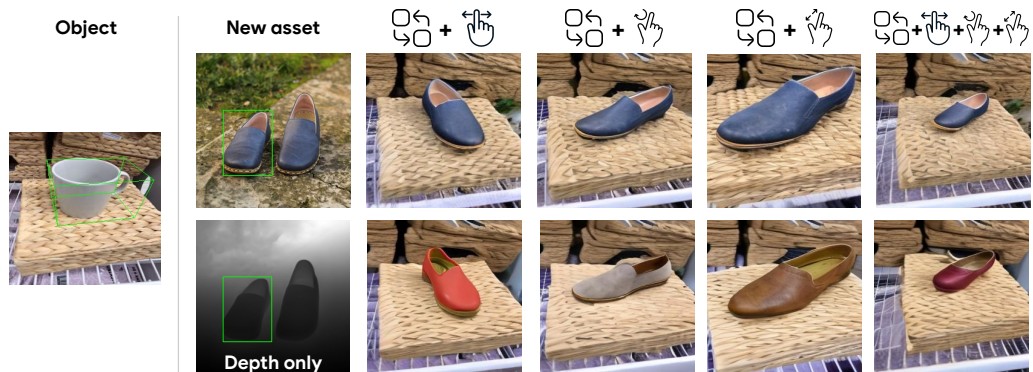

Figure 5: Object replacement examples with appearance and geometry conditioning (top) and geometry conditioning (bottom). Camera pose is fixed

- *MOVi-E* (Greff et al., 2022) is a synthetic generated using Blender scenes with up to 23 objects and consists of object and camera movement.
- *Objectron* (Ahmadyan et al., 2020) increases visual complexity by capturing single- and multi-object real world scenes. It consists of 15k videos of nine categories of objects.
- *Waymo Open* (Sun et al., 2020) is a dataset of real-world self-driving cars captured by a car-mounted camera from multiple angles.
- *EgoTracks* (Tang et al., 2023) is an annotated version of *Ego4D* (Grauman et al., 2022) consisting of 22.5k object tracks derived from 5.9k videos. The dataset contains a vast number of objects, many of which are only seen once, and challenging egocentric movement. Unlike the other datasets, EgoTracks only contains 2D bounding boxes.

Each of these datasets consists of video sequences of scenes containing various objects with 2D and 3D bounding box annotations. We filter out objects with small bounding boxes and randomly flip images. We list additional dataset information in section A.

### 4.1 POSE, APPEARANCE AND GEOMETRIC CONTROL

Neural USD exposes various ways for the user to interact with the image model that were previously unavailable. In Fig. 4, we demonstrate how Neural USD can let the user translate, rotate, and scale objects as desired. Additionally, users can choose to condition solely on geometry, which leads to novel appearances that satisfy the 3D structure of the original object. Neural USD also exposes the ability to replace objects with other desired objects, or to replace the background in an image. In Fig. 5, we show how Neural USD can be used to replace objects in a scene through geometric and appearance conditioning. In all of these examples, we see that other elements in the scene do not change. Additional editing examples can be found in section F.

### 4.2 ITERATIVE EDITING

Neural USD allows users to make multiple edits to a given image. Throughout the sequence of edits, other scene elements either do not change or change in ways that are 'continuation' of the original image. For example, new features of the background might be visible but the new features is consistent with the original background. Thus, our method does not make does not make unintended global changes. In Fig. 1, we showed how we can first change the pose of the object, then change the pose as well as the appearance and eventually all three elements (pose, appearance and geometry)

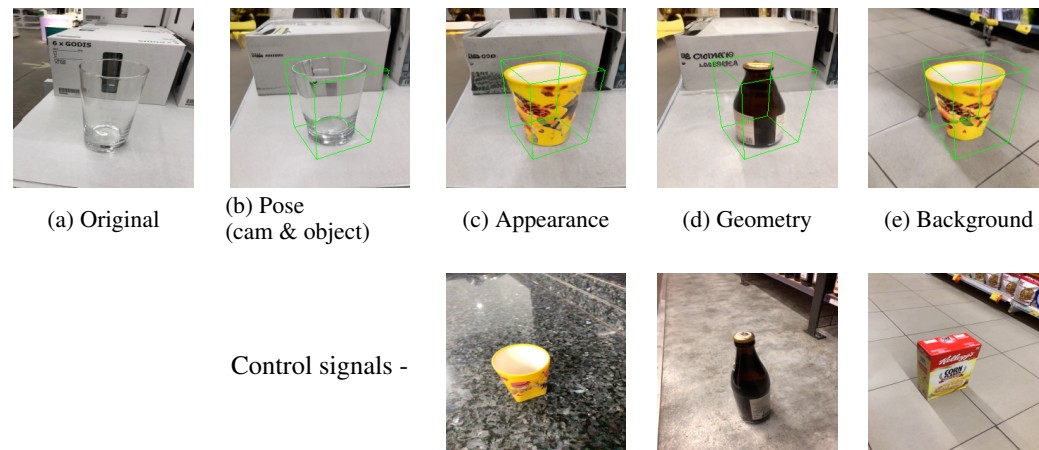

|  (a) Original | (b) Pose (cam & object) | (c) Appearance | (d) Geometry | (e) Background |

Control signals -

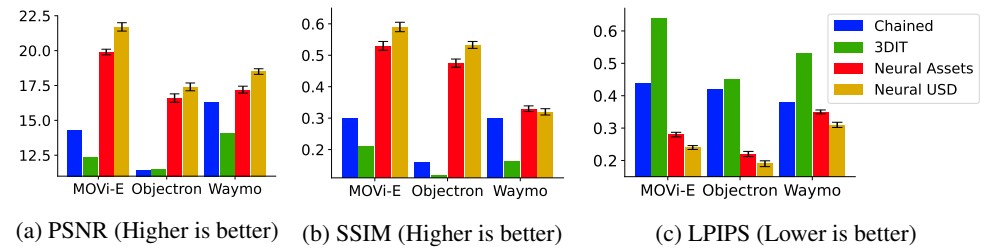

Figure 6: Given the original image (a), we specify the camera pose and desired target pose as a 3d bounding box (b). Next in (c), we condition the image to look like 'yellow cup'. In (d), we condition the original image to have the desired pose as in (b) and geometry and appearance like the bottle. Note that this corresponds to replacing the original glass with this new bottle; whereas in (c), the glass changed its appearance. In (e), we again edit the pose (as in b), appearance (as in c) and background (make the background same as the bottom image) all at the same time.

(a) PSNR (Higher is better)  (b) SSIM (Higher is better)  (c) LPIPS (Lower is better)

Figure 7: Object control performance on MOVi-E, Objectron, and Waymo. Values measure quality of target reconstruction for single and multi-object scenes.

at the same time. In Fig. 6, we show another example. In this case, we also give more details on the conditioning images. Even more examples are available in Section E. To the best of our knowledge, this is the first model/framework to allow for such precise object control and enable iterative workflows.

### 4.3 COMPARISON AGAINST OTHER MODELS

We evaluated neural USD and baselines using two different criteria. The first simply measures the model's ability to correctly reconstruct the target image from the provided source encodings and target pose. We use SSIM (Wang et al., 2004), LPIPS (Zhang et al., 2018), and PSNR. We use these criteria to compare the model to other 3D-aware image editing approaches. These include *Neural Assets* (Wu et al., 2024), *Object 3DIT* (Michel et al., 2023), and Chained (Wu et al., 2024). Object 3DIT is limited in its ability to render large viewpoint changes as it does not encode camera poses, while Neural Assets only supports 3D bounding box and RGB appearance conditioning.

Fig. 7 shows that Neural USD outperforms Object 3DIT, Chained and improves over Neural Assets while introducing a more flexible conditioning format with additional control inputs. For these experiments, we extract source and target frames from the dataset which contain multi-object changes and compare the model's ability to accurately reconstruct the target image.

Next, we compare our approach to non 3D-aware baselines that allow for modification to images using other conditioning signals such as text, geometry, or appearance. We include *Instruct-Pix2Pix* (Brooks et al., 2022), which uses text descriptions to modify a source image, *Control-*

*Net* (Zhang et al., 2023), which supports various control signals during image generation, *T2I-Adapter* (Mou et al., 2023), which learns various adapters to support additional spatial control signals, and *Stable Diffusion v2.1* (Rombach et al., 2022), a large pre-trained text-to-image model. Additionally, we include a VqVAE baseline, consisting of a convolutional encoder, finite scalar quantization (Mentzer et al., 2023), and a UViT (Hoogeboom et al., 2023) decoder, and trained with a multi-scale denoising diffusion loss (Hoogeboom et al., 2023). A more comprehensive discussion of baselines is included in section B.

We measure the performance of these methods along two axes: reconstruction and controllability. To determine the model's reconstruction performance, we supply it with all available conditioning signals extracted from a source image and measure the similarity between the source image and the model prediction. Controllability is measured by providing a control input that describes the difference between the source scene and the target scene, and measuring the similarity between the model prediction and the target image.

Fig. 8 shows that the Neural USD out-performs all the above models along these two axes. ControlNet, Stable Diffusion, T2I adapters, and VqVAE only expose reconstruction interfaces, and don't allow for object-centric or global edits of an input image. As such, measuring their controllability is challenging, since proposing modifications to input conditioning signals like depth, edge maps, or masks is non-trivial. Alternatively, InstructPix2Pix does provide an interface for image-level edits via text, but does not allow for reconstruction of an image from input conditioning signals. Approaches that allow for both include Object 3DIT, Neural Assets, and Neural USD.

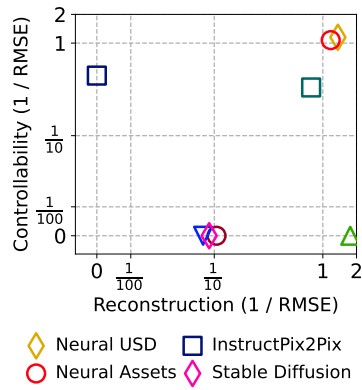

Figure 8: Reconstruction vs. Controllability performance on MOVi-E, Objectron, and Waymo. Axes are log-scale and values are reported as $^1/_{\text{RMSE}}$.

## 5 CONCLUSION

Neural USD introduces an object-centric conditioning framework for generative models, enabling precise control over appearance, geometry, and pose. Inspired by the Universal Scene Descriptor (USD), it encodes structured per-object attributes into conditioning vectors, ensuring cross-model compatibility. Using a fine-tuning approach with paired video frames, Neural USD disentangles control signals for independent object manipulation. Our framework enables a user to make multiple edits iteratively to the generated image such that other elements in the scene are consistent and do not change. Our experiments show superior performance in structured scene synthesis and object control; showcasing the potential of Neural USD as a flexible and portable standard for generative modeling.

In the future, we would like to scale our approach across various axes. Some of these include, training with a larger and more "modern" base model. Currently, we use StableDiffusion. Going forward, we would like to replace this with flux as this enables better visual quality. Our current results show the effectiveness of our approach on multiple datasets like Objectron, Waymo etc. Another possible direction is to scale our approach to much larger datasets.

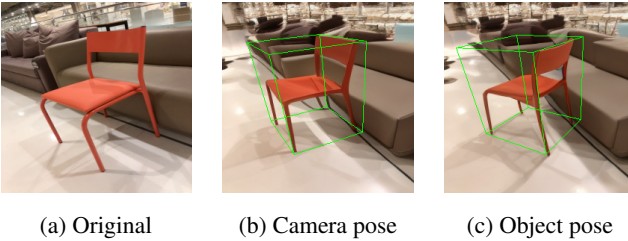

(a) Original      (b) Camera pose      (c) Object pose

Figure 9: Demo of iterative editing using the finetuned Neural USD model. Given the original image (a), we specify the camera pose (b). We then specify an additional object pose change (c).

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

# A    DATASETS

## A.1    EGOTRACKS

EgoTracks (Tang et al., 2023) is a tracked bounding box dataset consisiting of manually labeled 22.5k object tracks spanning 5.9k videos. Being a derivative dataset of Ego4D (Grauman et al., 2022), EgoTracks is ego-centric and features an extreme amount of foreground and background movement. Additionally, Ego4D object tracks often feature very small bounding boxes (e.g. for utensils). We filter the data in two ways: first, we filter out bounding boxes with small height or width, the threshold being 1/10th the normalized height and width of the screen. To prevent object tracks from leaving the field of view, we sample source and target frames from within 15 frames of each other, and discard samples for which no object is present. Finally, during evaluation, we filter out results with motion blur or extreme background shift.

## A.2    OBJECTRON

Objectron (Ahmadyan et al., 2020) consists of 15,000 object-centric video clips featuring everyday objects across nine categories. Each video includes object pose tracking, allowing us to extract 3D bounding boxes. Since the dataset lacks 2D bounding box annotations, we generate them by projecting the eight corners of the 3D boxes onto the image and computing the tightest bounding box around the projected points.

## A.3    WAYMO OPEN

Waymo Open (Sun et al., 2020) comprises 1,000 video clips of self-driving scenes captured by car-mounted cameras. Following previous studies (Wu et al., 2024), we use the front-view camera and car bounding box annotations. The 3D bounding boxes include only the heading angle (yaw-axis rotation), so we set the other two rotation angles to zero. Additionally, the provided 2D and 3D boxes are misaligned, making paired frame training unfeasible. To address this, we project the 3D boxes to obtain corresponding 2D boxes, similar to the approach used for Objectron.

## A.4    MOVI-E

MOVi-E (Greff et al., 2022) includes 10,000 videos simulated using Kubric (Greff et al., 2022), with each scene featuring 11 to 23 real-world objects from the Google Scanned Objects (GSO) repository (Downs et al., 2022). At the beginning of each video, multiple objects are dropped onto the ground, causing them to collide. The scene's lighting comes from a randomly sampled environment map. The camera follows a simple linear motion.

# B  BASELINES

## B.1  OBJECT 3DIT

Object 3DIT (Michel et al., 2023) fine-tunes Zero-1-to-3 (Liu et al., 2023a) for scene-level 3D object editing. We derive editing instructions from the target object pose, including translation and rotation. However, this lacks support for significant viewpoint changes as it does not encode camera poses. We use the official code and pre-trained weights of the Multitask variant.

## B.2  INSTRUCTPIX2PIX

InstructPix2Pix enables text-guided image editing by fine-tuning a diffusion model to follow editing instructions. It conditions on both an input image and a text prompt, learning to predict pixel changes based on the instruction. However, it lacks explicit 3D control and struggles with complex multi-object edits. We construct a dataset of 100 source target pairs and their differences describes as text prompts. InstructPix2Pix is then conditioned on the source image and text prompt, and we evaluate how accurate it is at reconstructing the target image. We find that the model struggles to elicit the fine changes in object pose described in the text.

## B.3  T2I ADAPTERS

T2I-Adapters (Mou et al., 2023) enable additional conditioning mechanisms for pre-trained diffusion models, allowing control beyond text prompts. They integrate spatial signals like depth maps or segmentation masks to guide image generation while preserving the original model's structure. These adapters typically introduce lightweight modules, such as attention layers or zero-initialized convolutions, that fuse external control signals with the model's latent space. We condition the T2I-adapters model on the spatial modalities corresponding to the source image, such as masks and depth, and evaluate its performance in reconstructing the source image.

## B.4  NEURAL ASSETS

Neural Assets (Wu et al., 2024) introduces a per-object representation for 3D-aware multi-object control in image diffusion models. It encodes appearance and pose separately, allowing object manipulation, including translation, rotation, and rescaling. We evaluate Neural Assets using the same criteria used for Neural USD: we extract source modalities from a source image and condition on the target poses derived from the target image. We then measure the reconstruction loss with the target image.

## B.5  CONTROLNET

ControlNet (Zhang et al., 2023) enables spatial conditioning in diffusion models by introducing trainable layers that process external control signals, such as edge maps, depth maps, or pose keypoints. It retains the original model's weights while adding zero-initialized convolution layers. This allows for control over image generation. However, it does not allow for object-centric image editing, as changes to the conditioning signals can lead to global changes in the image. We condition the Control model on the spatial modalities corresponding to the source image, such as masks and depth, and evaluate its performance in reconstructing the source image.

## C  HYPERPARAMETERS

Here we outline the hyperparameters used to implement and train Neural USD.

Table 1: Hyperparameters for Neural USD.

| PARAMETER | VALUE |
|---|---|
| STABLE DIFFUSION VARIANT | v2.1 |
| DINO VARIANT | VIT-B/8 |
| DINO FEATURE MAP SIZE | $28 \times 28$ |
| INPUT IMAGE SIZE | $256 \times 256$ |
| TOKEN DIMENSION | 1024 |
| BATCH SIZE | 512 |
| OPTIMIZER | ADAM |
| STABLE DIFFUSION LR | $1 \times 10^{-4}$ |
| IMAGE ENCODER (DINO) LR | $5 \times 10^{-4}$ |
| WARMUP STEPS | 2000 |
| DECAY SCHEDULE | LINEAR |
| FINE-TUNING STEPS | 50000 |
| GRADIENT CLIP VALUE | 1.0 |
| MODALITY DROPOUT PROBABILITY | 0.25 |
| POSE DROPOUT PROBABILITY | 0.25 |
| ALL CONDITIONING DROPOUT PROBABILITY | 0.1 |
| CFG WEIGHT | 3.0 |

# D   QUALITATIVE BASELINES

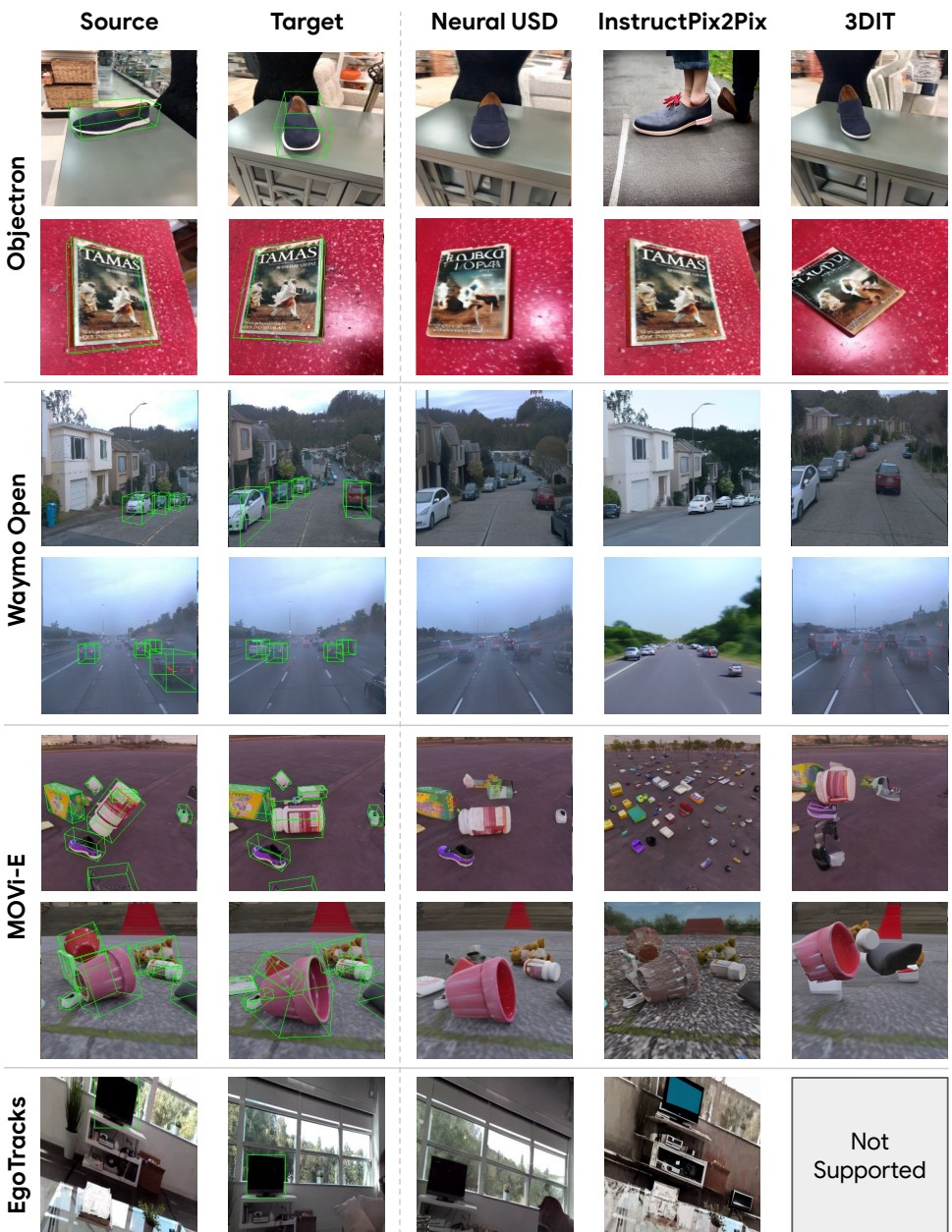

Figure 10: Object pose conditioning performance on MOVi-E, Objectron, Waymo Open, and Ego-Tracks. Models generate the target image provided a source image and the 3D bounding box targets (Neural USD, 3DIT) or textual prompts (InstructPix2Pix). Our method satisfies the desired pose while preserving the foreground and background appearance. InstructPix2Pix fails to elicit object movement.

# E    ITERATIVE WORKFLOW EXAMPLES

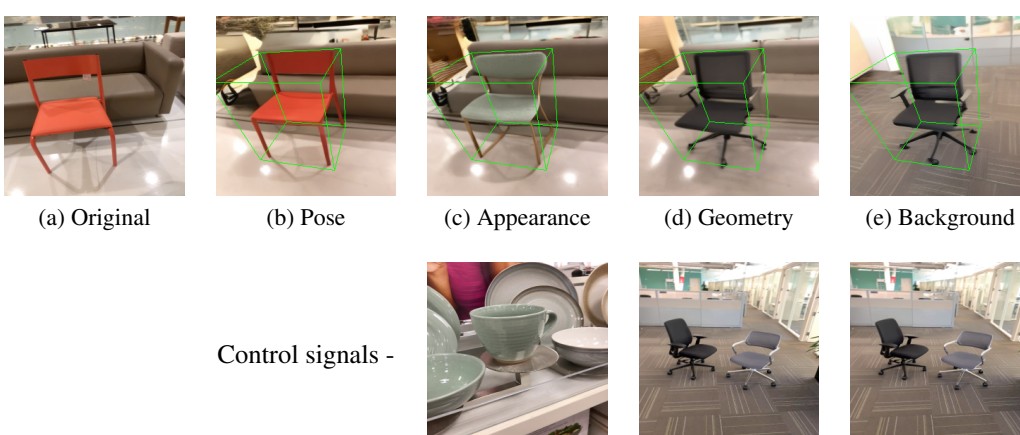

|  |  |  |  |  |
|--|--|--|--|--|
| (a) Original | (b) Pose | (c) Appearance | (d) Geometry | (e) Background |

Control signals -

Figure 11: Given the original image (a), we first change the pose of the chair as desired in (b). Next in (c), we condition the original image to look like the cup while also being in the new pose (as in b). In (d), we condition the original image to have the desired pose (as in b) and geometry and appearance like the black chair. In (e), we further ask the model to change the background to be as the bottom image. Note that in this example, we use different aspects of the same image for both geometric and background conditioning.

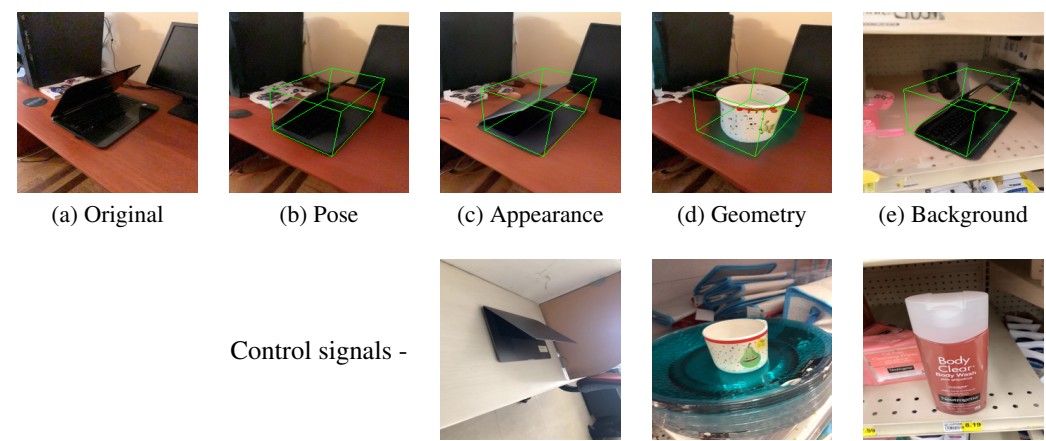

|  |  |  |  |  |
|--|--|--|--|--|
| (a) Original | (b) Pose | (c) Appearance | (d) Geometry | (e) Background |

Control signals -

Figure 12: We first change the pose (b). Next in (c), we condition the original image to look as specified. In (d), we replace the laptop with another object. In (e), we edit the pose (as in b) and background (make the background same as the bottom image).

# F  ADDITIONAL EXPERIMENTAL RESULTS

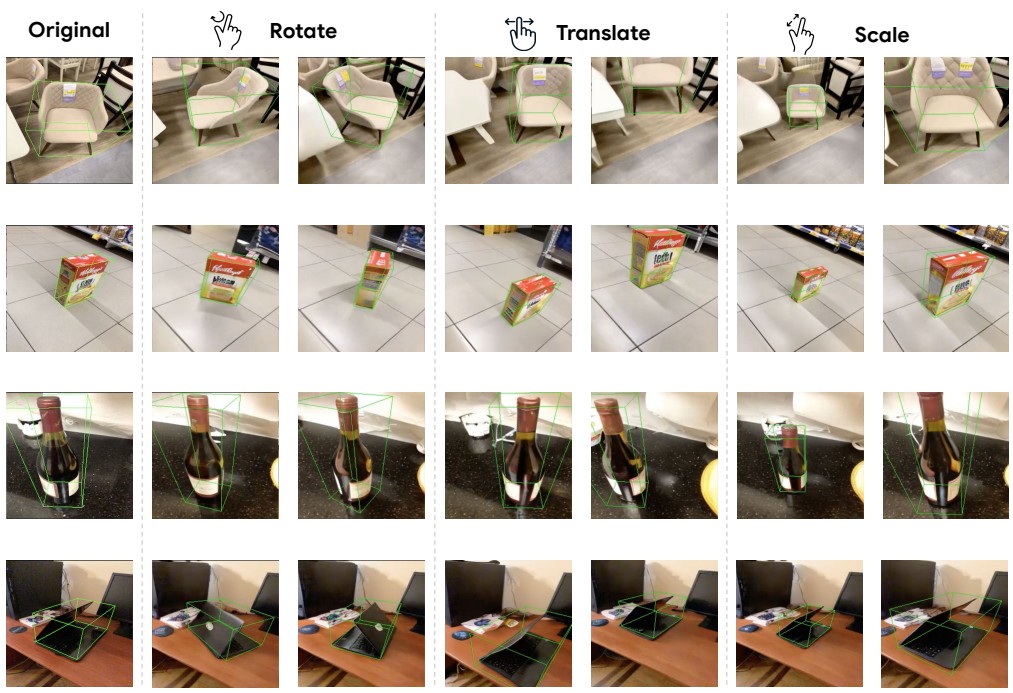

Figure 13: Additional Objectron 3D pose control examples. Camera pose is fixed

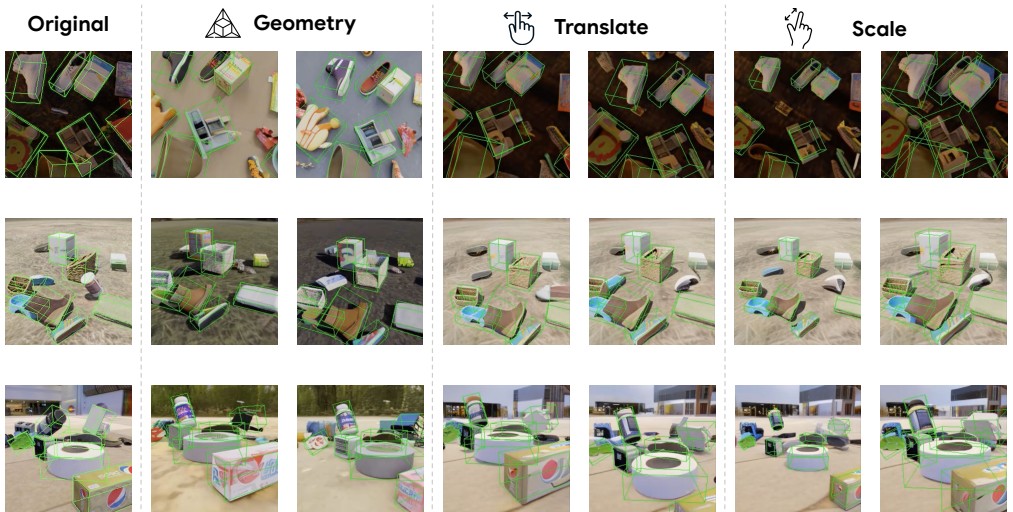

Figure 14: Additional MOVi-E 3D pose control examples. Camera pose is fixed.

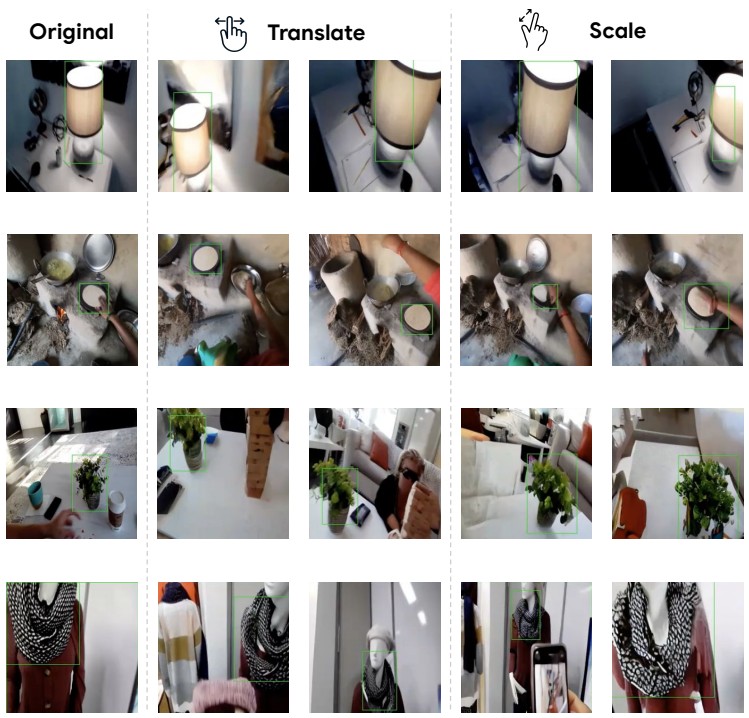

Figure 15: Additional Egotracks 2D bounding box control examples.

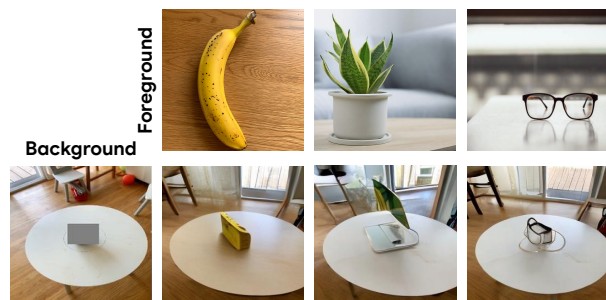

Figure 16: When trained on limited data - a handful of labeled datasets constituting ¡50,000 sequences - Neural USD fails to generalize to new object categories. This lack of generalization can likely be remedied by co-training on more readily-available 2D bounding box datasets.

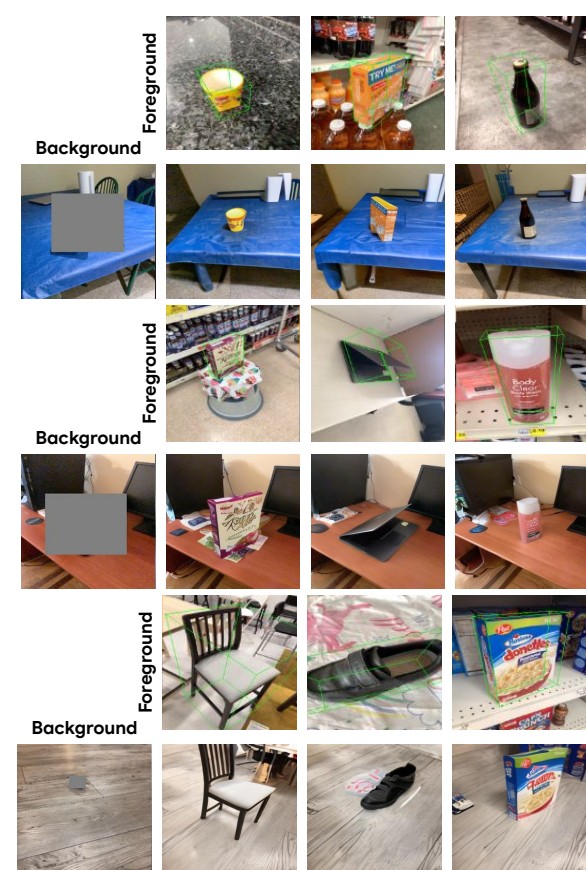

Figure 17: Foreground background replacement. Neural USD allows for easy swapping of assets in the scene. The background, simply being another asset, can be replaced with reference modalities.

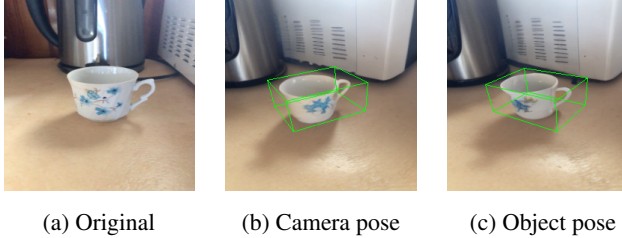

(a) Original          (b) Camera pose          (c) Object pose

Figure 18: Demo of iterative editing using the finetuned Neural USD model. Given the original image (a), we specify the camera pose (b). We then specify an additional object pose change (c).

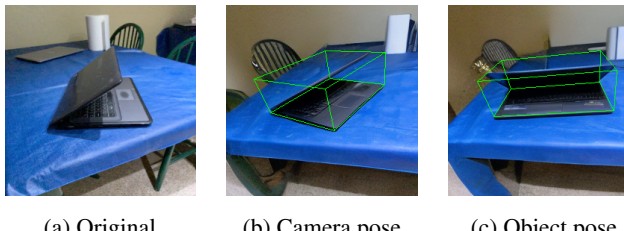

(a) Original      (b) Camera pose      (c) Object pose

Figure 19: Demo of iterative editing using the finetuned Neural USD model. Given the original image (a), we specify the camera pose (b). We then specify an additional object pose change (c).

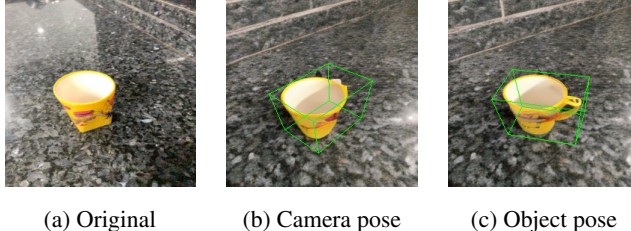

(a) Original      (b) Camera pose      (c) Object pose

Figure 20: Demo of iterative editing using the finetuned Neural USD model. Given the original image (a), we specify the camera pose (b). We then specify an additional object pose change (c).

