# OpenReview forum: "Neural USD: An object-centric framework for iterative editing and control"
_ICLR.cc/2026/Conference — Submitted to ICLR 2026_

### Official Review · Reviewer_sNZo · 2025-10-29

**Soundness:** 2
**Presentation:** 2
**Contribution:** 2
**Rating:** 2
**Confidence:** 4

**Summary:**

This paper introduces "Neural USD," an object-centric framework designed to enable precise and iterative editing of generative models. Taking inspiration from the Universal Scene Descriptor (USD) standard in computer graphics, the authors propose representing a scene's components (objects, background) as assets with distinct, hierarchical attributes: appearance, geometry, and pose. The core technical contribution is a fine-tuning strategy that uses paired images from video sequences to train a generative model to disentangle these control signals. The paper demonstrates that this framework allows for object-level manipulations, such as changing pose, appearance, or geometry, and replacing objects or backgrounds, while aiming to keep other scene elements consistent.

**Strengths:**

Clever Conceptual Bridging: The core idea of adapting the structured, hierarchical USD standard from computer graphics to the conditioning of diffusion models is both novel and elegant. It provides a principled-sounding approach to a problem often tackled with less structured methods.

Core Training Strategy: The method of using paired images ($I_{src}$ for appearance/geometry, $I_{tgt}$ for pose) to force the model to learn disentangled representations is a key insight and a strong methodological contribution.

**Weaknesses:**

The paper's primary qualitative example, Figure 1, seems to undermine its central claim of disentangled control. In Fig 1(b), the stated operation is a "Pose" change. However, the background has clearly changed in appearance compared to Fig 1(a). Furthermore, the object's own appearance (the orange chair) also appears to have different lighting/shading in 1(b).

Missing Critical Ablation Studies (Especially on Geometry): The framework's complexity (requiring pose, appearance, and geometry) is not sufficiently justified. The authors have not provided a crucial ablation study to demonstrate the necessity of the geometry (e.g., depth map) signal. How does the model perform with only Pose + Appearance conditioning? As user paste the warped cropped region into background, then inpaint the image, it seems to get the similar result? Without this ablation, it is impossible to assess the contribution of the geometry component. This is a significant gap in the experimental validation.

(Minor Weakness): As noted, the main paper is surprisingly sparse on qualitative results, relegating most examples to the appendix.  Given that the paper does not fill the 9-page, key supporting results (especially those that successfully demonstrate the claims from Fig 1) should have been included in the main body.

**Questions:**

Same as weakenss.

---

> ### Author Response · Authors · 2025-11-24
> **Response to reviewer sNZo**
>
> We thank the reviewer for their detailed review, and for noting the “Clever Conceptual Bridging” of Neural USD as it related to computer graphics, and for noting that our training strategy is a “key insight” and a “strong methodological contribution”. We address the reviewer’s concerns below:
>
> # Addressing Weaknesses
>
> ## Disentangled control concerns
>
> We thank the reviewer for noting this issue, and for prompting us to take a closer look at our visualization code. Upon further inspection, we discovered the presence of an obscure bug that snuck into our visualization pipeline in the later stages of creating figures for the paper. This bug caused *both* object *and* camera pose to change whenever the object pose was modified, but only for *some* examples. This is clear from looking at the examples in the paper:
>
> * In figures 1 and 6 (*after* bug introduced):
>   * In (a), we make edits to both the object *and* camera pose \- leading to background changes due to camera movement.
>   * In (c, d), we make significant modifications to the object appearance and geometry, and the background remains largely unchanged.
> * In figures 4, 5, 12, and 13 (*before* bug introduced):
>   * The translation and rotation examples introduce significant modifications to the object’s foreground pose. However, **the background remains unchanged as desired**.
>
> We have included two additional figures in the appendix and modified the captions for figures 1 and 6 to highlight that these contain *both* object and camera pose modifications. We again thank the reviewer for spotting this bug. We also want to mention that this bug *does not* affect any of the metrics reported in the paper, as it was strictly a visualization bug.
>
> **Re: Variations in lighting/shading.** Some level of lighting/shading differences are due to the variance of lighting directions in the dataset. Since lighting is not a conditioning signal, the model outputs lighting and shadows that are most consistent with the input signal.
>
> ## Clarifying the use of geometry as a conditioning signal
>
> Our goal in introducing geometry as a conditioning signal is not to claim that depth inherently improves reconstruction quality, but to expose a *distinct axis of controllability* that is otherwise unattainable. Geometry provides structural information—object shape, volume, and spatial layout—that neither appearance nor pose alone can express. Prior work has not demonstrated independent and compositional control across RGB appearance, pose, and geometric structure within a single unified framework. Neural USD is the first to support arbitrary combinations of these modalities, enabling operations such as geometry-preserving object replacement, shape-consistent editing, or cross-object geometry transfer (e.g., Fig. 5), which cannot be reproduced by using only pose and appearance. The contribution of geometry is therefore architectural and functional: it unlocks a new mode of object-level manipulation rather than serving as a quality booster for the other modalities.
>
> ## Paper Formatting
>
> We have modified the manuscript to bring more examples into the main portion of the paper and showcase more visual results. We thank the reviewer for this suggestion.
>
> # Request for clarification for the reviewer
>
> > As user paste the warped cropped region into background, then inpaint the image, it seems to get the similar result?
>
> Could the reviewer kindly elaborate which example they are referring to here? Neural USD doesn’t “paste” images into the background, it conditions the image model on per-object modalities. The background is treated as a global “scene” entity, and has its own corresponding appearance and geometry embeddings. Clarification is kindly requested to determine which additional changes to make.
>
> # Manuscript Changes
>
> We again thank the reviewer for their insightful feedback. Their comments have led to the following revisions:
>
> * Discovered a visualization bug which led to unintended modifications to both camera and object pose in some figures. Figure captions were updated to highlight that these examples modify both camera and object pose.
> * Added two new examples highlighting the Neural USD’s ability to disentangle object and camera pose edits.
> * Re-formatted the paper to bring more visual examples into the main text.
>
> ---
>
> *We hope our responses have fully clarified the contributions and addressed the reviewer’s concerns (mainly the bug pertaining to undesired background edits). If so, we would be grateful if the reviewer would consider updating their score.*

---

> ### Comment · Reviewer_sNZo · 2025-11-27
> **Comment to authors**
>
> Thank you for your detailed response. However, after carefully checking your rebuttal and the (presumably) revised manuscript, I have decided to maintain my original score. My concerns regarding the validity of the disentanglement and the necessity of the proposed components remain unresolved.
>
> 1. **Discrepancy regarding the Revised Manuscript.** I carefully checked the uploaded PDF, but I could not find the updates mentioned in your rebuttal.  Please verify if the correct version of the PDF was uploaded. As it stands, I cannot evaluate the "new examples" or "re-formatting" you claimed to have implemented.
>
> 2. Persistent Issues in Figure 1 (Disentanglement).  Regarding the "visualization bug" explanation: While I understand your claim that the camera pose shifted, this does not fully explain the visual artifacts. Observation: In Figure 1, beyond the perspective shift, the color of the background sofa and the details in the top-left corner show significant inconsistency between (a), (b), and (c). A mere change in camera pose should not drastically alter the shading or color tone of static background objects if the appearance control is truly disentangled. The visual evidence suggests that the model is still hallucinating or leaking appearance information, rather than performing a clean geometric transformation.
>
> 3. Missing Ablation Study & Clarification on "Paste and Inpaint". You asked for clarification on my comment: "As user paste the warped cropped region into background, then inpaint the image, it seems to get the similar result?"
>
> I am referring to a trivial baseline. In examples like Figure 3 (changing shoes), one could simply crop the target shoe, paste it onto the background, and use a standard diffusion inpainting model to blend the boundaries. The result would likely be visually similar to your output.
>
> I have re-examined the entire paper structure, including Section 4 and the Appendix. The experiments only compare Neural USD against external baselines.  There is no ablation study verifying the contribution of individual internal components (e.g., Neural USD w/o Geometry vs. Full Model).
>
> As I noted, for simple tasks like object replacement (Figure 3), a trivial "crop-paste-inpaint" workflow could likely achieve similar results. Without an ablation study showing that the Geometry signal adds value beyond simple Appearance+Pose conditioning, the technical contribution stands unverified.

---

> > ### Author Response · Authors · 2025-12-03
> > **Response to reviewer sNZo**
> >
> > We are happy to share that we have added **four additional figures** (Figures 9, 18, 19, 20) to the revised manuscript, which we believe fully address concerns (1) and (2). These additional experiments showcase the model's ability to disentangle camera and object pose specification. We showcase this ability by iteratively editing first the camera pose, then the object pose across four different scenes.
> >
> > In regards to the requested experiment (3), we do not believe this is a valid comparison to Neural USD, as it would not allow for rotating or translating the "pasted image" in 3D, and would not support partial specification of appearance or depth.
> >
> > ---
> >
> > *We again thank the author for their feedback, and encourage them to review the revised manuscript, which addresses the concerns they had with camera/object pose disentanglement.*

---

### Official Review · Reviewer_PGAs · 2025-10-31

**Soundness:** 3
**Presentation:** 3
**Contribution:** 3
**Rating:** 8
**Confidence:** 4

**Summary:**

This paper proposes Neural USD. In general, it can be seen as a ControlNet with a finer grid for object control. The paper generates a dataset with detailed annotations such as depth, boxes, and many others, then uses this information to fine-tune a pre-trained image model. The fine-tuned model shows great results in image control capability.

**Strengths:**

1. The paper targets a great problem for finer control of the objects in an image. The proposed method, although simple, is pretty straightforward and effective.

2. The datasets in the paper can benefit future research.

3. The demonstrated results are good.

**Weaknesses:**

1. Since the model is still a learning-based image-to-image model, keeping other objects unchanged is not guaranteed. I can see obvious background change in Figure 1.

**Questions:**

I do not see major weaknesses or questions.

---

> ### Author Response · Authors · 2025-11-24
> **Response to reviewer PGAs**
>
> We thank the reviewer for thoughtful comments, and for recognizing Neural USD’s presentation, experiments, and the simple and straightforward effectiveness of our method. We address the reviewer’s concerns below:
>
> # Addressing Weaknesses
>
> ## Unwanted global scene changes
>
> We thank the reviewer for noting this issue, and for prompting us to take a closer look at our visualization code. Upon further inspection, we discovered the presence of an obscure bug that snuck into our visualization pipeline in the later stages of creating figures for the paper. This bug caused *both* object *and* camera pose to change whenever the object pose was modified, but only for *some* examples. This is clear from looking at the examples in the paper:
>
> * In figures 1 and 6 (*after* bug introduced):
>   * In (a), we make edits to both the object *and* camera pose \- leading to background changes due to camera movement.
>   * In (c, d), we make significant modifications to the object appearance and geometry, and the background remains largely unchanged.
> * In figures 4, 5, 12, and 13 (*before* bug introduced):
>   * The translation and rotation examples introduce significant modifications to the object’s foreground pose. However, **the background remains unchanged as desired**.
>
> We have included two additional figures in the appendix and modified the captions for figures 1 and 6 to highlight that these contain *both* object and camera pose modifications. We again thank the reviewer for spotting this bug. We also want to mention that this bug *does not* affect any of the metrics reported in the paper, as it was strictly a visualization bug.
>
> # Manuscript Changes
>
> We again thank the reviewer for their insightful feedback. Their comments have led to the following revisions:
>
> * Discovered a visualization bug which led to unintended modifications to both camera and object pose in some figures. Figure captions were updated to highlight that these examples modify both camera and object pose.
> * Added two new examples highlighting the Neural USD’s ability to disentangle object and camera pose edits.
>
> ---
>
> *We hope our responses have fully clarified the contributions and addressed the reviewer’s concerns. If so, we would be grateful if the reviewer would consider updating their score.*

---

### Official Review · Reviewer_WFwE · 2025-11-01

**Soundness:** 3
**Presentation:** 3
**Contribution:** 3
**Rating:** 6
**Confidence:** 3

**Summary:**

This work introduces a new object-centric generative framework inspired by USD standard, aiming to unify representation, control, and editing in image and 3D content generation. The core idea of this method is to tokenize each object in a scene, capturing its geometry, appearance, pose, and material attributes, into a structured latent representation that can condition diffusion or transformer-based generative models.

**Strengths:**

1. The idea is interesting to me, that is a unified, structured conditioning standard inspired by USD, enabling disentangled control over object appearance, geometry, and pose in generative models.
2. The ability to perform multi-step, fine-grained, object-level edits without unintended global changes, a clear improvement over existing conditioning methods (e.g., ControlNet, InstructPix2Pix).
3. The format is architecture-agnostic and supports diffusion, DiT, and transformer models through tokenized conditioning, improving portability and generalizability.

**Weaknesses:**

1. Global scene changes still occur, as I observed from the visual results. It is somewhat overclaimed, as the abstract and introduction sections stated.
2. It remains unclear whether the fusion happens at the feature level (joint embedding) or via concatenated conditioning channels.
3. Uses Stable Diffusion v2.1 as backbone, leading to lower image quality than state-of-the-art diffusion models; I still believe it should be adopted in more powerful backbones (e.g., Flux).

**Questions:**

Please refer to the weakness part

---

> ### Author Response · Authors · 2025-11-24
> **Response to reviewer WFwE**
>
> We thank the reviewer for noting the soundness, presentation, and contributions of Neural USD \- and for noting the value of Neural USD as a unified, structured representation and its ability to support iterative workflows with an architecture-agnostic approach. We address concerns below:
>
> # Addressing Weaknesses
>
> ## 1\. Unwanted global scene changes
>
> We thank the reviewer for noting this issue, and for prompting us to take a closer look at our visualization code. Upon further inspection, we discovered the presence of an obscure bug that snuck into our visualization pipeline in the later stages of creating figures for the paper. This bug caused *both* object *and* camera pose to change whenever the object pose was modified, but only for *some* examples. This is clear from looking at the examples in the paper:
>
> * In figures 1 and 6 (*after* bug introduced):
>   * In (a), we make edits to both the object *and* camera pose \- leading to background changes due to camera movement.
>   * In (c, d), we make significant modifications to the object appearance and geometry, and the background remains largely unchanged.
> * In figures 4, 5, 12, and 13 (*before* bug introduced):
>   * The translation and rotation examples introduce significant modifications to the object’s foreground pose. However, **the background remains unchanged as desired**.
>
> We have included two additional figures in the appendix and modified the captions for figures 1 and 6 to highlight that these contain *both* object and camera pose modifications. We again thank the reviewer for spotting this issue. We also want to mention that this bug *does not* affect any of the metrics reported in the paper, as it was strictly a visualization bug.
>
> ## 2\. Latent fusion clarification
>
> We apologize for not making this clearer in the paper, and will update the manuscript to reflect the following clarifications: After encoding, each conditioning modality is fused by concatenating its channels. This conditioning signal is then used as the control for the image model.
>
> ## 3\. Choice of backbone
>
> We agree that a more powerful image generation base model, like Flux, would have resulted in visually more stunning results. Unfortunately adopting this backbone was out of scope for this project for operational reasons, but we believe that this backbone choice is mostly orthogonal to our scientific contribution.
>
> # Manuscript Changes
>
> We again thank the reviewer for their insightful feedback. Their comments have led to the following revisions:
>
> * Discovered a visualization bug which led to unintended modifications to both camera and object pose in some figures. Figure captions were updated to highlight that these examples modify both camera and object pose.
> * Added two new examples highlighting the Neural USD’s ability to disentangle object and camera pose edits.
> * Revised wording in the paper to better explain how conditioning signals are fused.
>
> ---
>
> *We hope our responses have fully clarified the contributions and addressed the reviewer’s concerns. If so, we would be grateful if the reviewer would consider updating their score.*

---

### Official Review · Reviewer_Mwtn · 2025-11-03

**Soundness:** 3
**Presentation:** 2
**Contribution:** 2
**Rating:** 4
**Confidence:** 3

**Summary:**

This work enables precise control over an object’s appearance, geometry, and pose. By encoding geometry, depth maps, and bounding boxes into the image generation model, it can control the placement of specific objects within specific scenes, as well as edit their pose/appearance, background, and viewpoint. Moreover, it supports an iterative editing workflow, allowing continuous refinement of scene composition and object attributes.

**Strengths:**

1.	Achieves disentangled control over an object’s appearance and geometry in a simple and straightforward manner.
2.	Enables more precise control of objects, with quantitative metrics outperforming previous methods.
3.	Designs a stable iterative 3D editing workflow, allowing sequential replacement of pose, appearance, object, and background, while preserving the results of previous edits during new editing steps.

**Weaknesses:**

1.	The paper repeatedly emphasizes that it can edit an object’s pose while keeping other attributes of the source image unchanged (as stated in the abstract, section 4.2). However, in practice, the camera pose and object pose are not successfully disentangled — pose editing often causes a change in the viewing angle instead of the object moving relative to the scene (see Fig. 9).
2.	This work follows a technical route very similar to Neural Assets, with comparable metrics (see Figs. 7 and 8), yet lacks corresponding visual comparisons.
3.	The background after editing differs noticeably from the original image (Figs. 5 and 6), and the object texture and appearance exhibit clear artifacts (e.g., Fig. 6 (c)). The authors attribute this to the base image model not being state-of-the-art, but this explanation remains unverified. The presence of such visible artifacts after even a single edit undermines the benefit of an iterative workflow, since preventing cumulative errors across edits is one of its key goals.
4.	The contributions are relatively limited.
Contribution 1 claims that the model learns disentangled control signals from video image pairs, but this training paradigm was already introduced by Neural Assets. The main difference lies in the additional encoding of geometry, enabling finer-grained disentanglement.
Discussion of Contribution 2 is provided in the previous point 3.
Contribution 3 claims improved accuracy based on Fig. 8, but the figure shows no significant improvement over Neural Assets.
5.	In Figs. 5 and 6, the 3D bounding boxes before and after editing should be marked.

**Questions:**

See the weakness.

---

> ### Author Response · Authors · 2025-11-24
> **Response to reviewer Mwtn (1/2)**
>
> We thank the reviewer for their insightful comments, and for noting Neural USD’s sound presentation and ability to achieve precise, disentangled control of object’s appearance and geometry in a simple and straightforward manner \- and for engaging constructively with our work. We address the concerns below:
>
> # Addressing Weaknesses
>
> ## (1 & 3a) Disentanglement concerns and background
>
> We thank the reviewer for noting this issue, and for prompting us to take a closer look at our visualization code. Upon further inspection, we discovered the presence of an obscure bug that snuck into our visualization pipeline in the later stages of creating figures for the paper. This bug caused *both* object *and* camera pose to change whenever the object pose was modified, but only for *some* examples. This is clear from looking at the examples in the paper:
>
> * In figures 1 and 6 (*after* bug introduced):
>   * In (a), we make edits to both the object *and* camera pose \- leading to background changes due to camera movement.
>   * In (c, d), we make significant modifications to the object appearance and geometry, and the background remains largely unchanged.
> * In figures 4, 5, 12, and 13 (*before* bug introduced):
>   * The translation and rotation examples introduce significant modifications to the object’s foreground pose. However, **the background remains unchanged as desired**.
>
> We have included two additional figures in the appendix and modified the captions for figures 1 and 6 to highlight that these contain *both* object and camera pose modifications. We again thank the reviewer for spotting this bug. We also want to mention that this bug *does not* affect any of the metrics reported in the paper, as it was strictly a visualization bug.
>
> ## 2\. Neural Assets Comparisons
>
> We agree that Neural USD and Neural Assets share features in common \- namely the use of *per-object* conditioning signals to achieve object-level control, and in using video frames to disentangle pose representations. However, we argue that *Neural USD generalizes Neural Assets* by offering a wider range of control over object attributes, and by introducing an *iterative workflow* that enables an iterative, incremental composition of scenes.
>
> We *do not* expect Neural USD to outperform Neural Assets in terms of visual comparison for the modalities and control flow that Neural Assets supports. However, Neural USD does offer more control over object attributes and is more amenable to an iterative workflow.
>
> ## 3b. Cumulative Errors
>
> As mentioned in *(1 & 3a) Disentanglement concerns and background*, the *undesired* background edits were due to modifications applied to *both* the object and camera pose (added new figures). Neural Assets *can* handle iterative editing, precisely because it does not project back into pixel (or latent space) between edits. Instead, all information is held in the Neural USD, and image edits are tied only to Neural USD modifications.
>
> ## 4\. Limited Contributions and Comparison to Neural Assets
>
> We wish to highlight how Neural USD *generalizes* while building upon Neural Assets:
>
> * Neural USD provides a recipe for supporting arbitrary input modalities for conditioning. Whereas Neural Assets only supports image and pose input signals, we present a recipe that allows Image models to condition on far more conditioning signals, demonstrated at the example of depth/geometry conditioning, and study the implementation details that support the use of more modalities.
> * While Neural Assets only showcased 3D pose conditioning examples on small datasets (e.g. Objectron, Waymo Open) and relied on relative pose transformation between cameras as a conditioning signal, we demonstrate 2D pose conditioning examples on the much broader Ego4D dataset without the use of relative pose transformations, which are infeasible to acquire for most real-world data.
> * Neural Assets proposes individual edits to scene objects, whereas Neural USD introduces a more general *iterative workflow* for object editing.
> * Finally, we introduce a novel controllability v.s. reconstruction metric, which we argue is the proper way to benchmark conditioning formats for image generation models.
>
> We also express that any “improved accuracy” over Neural Assets is marginal, and we have humbly removed that statement from the manuscript. Instead, Neural USD offers greater control over object attributes, and an iterative workflow for incremental, iterative editing of scenes.
>
> ## 5\. Including bounding boxes after editing
>
> We apologize for the lack of bounding boxes after editing in figures 5 and 6\. We aimed for those figures to have a “cleaner” appearance (without additional annotations). The rest of the figures in the paper all include bounding boxes, and the additional figures also have bounding boxes.

---

> > ### Author Response · Authors · 2025-11-24
> > **Response to reviewer Mwtn (2/2)**
> >
> > # Manuscript Changes
> >
> > We again thank the reviewer for their insightful feedback. Their comments have led to the following revisions:
> >
> > * Discovered a visualization bug which led to unintended modifications to both camera and object pose in some figures. Figure captions were updated to highlight that these examples modify both camera and object pose.
> > * Added two new examples highlighting the Neural USD’s ability to disentangle object and camera pose edits.
> > * Revised wording in the paper to better put our contributions in context to Neural Assets.
> >
> > ---
> >
> > *We hope our responses have fully clarified the contributions and addressed the reviewer’s concerns. If so, we would be grateful if the reviewer would consider updating their score.*

---

### Author Response · Authors · 2025-12-03
**Summary of Rebuttal and Responses to Reviewers**

We thank all reviewers for their constructive feedback. Across the rebuttal, we focused on clarifying disentanglement behavior, addressing qualitative inconsistencies, explaining the role of geometry, and updating the manuscript to reflect corrections and new experiments.

# Reviewer Mwtn

**Main concerns:**
 (1) Apparent entanglement of object and camera pose; (2) Missing visual comparisons to Neural Assets; (3) Background drift and artifacts undermining iterative editing; (4) Limited contribution relative to Neural Assets; (5) Missing bounding-box annotations.

**Our response:**
 – We identified a visualization-only bug that unintentionally changed both camera and object pose in some examples. This explained background changes in Figures 1 and 6\. We emphasized that metrics were unaffected.
 – We added new figures demonstrating correct disentanglement and updated captions clarifying when both pose signals were modified.
 – We clarified the relationship to Neural Assets and reframed Neural USD’s contributions around (i) generalization to additional modalities, (ii) support for large-scale 2D pose conditioning, (iii) enabling iterative workflows, and (iv) proposing a new controllability-vs-reconstruction metric.
 – We removed the claim of improved accuracy and repositioned the contribution as broader controllability.
 – We acknowledged the missing bounding boxes and kept them in all other figures.

# Reviewer WFwE

**Main concerns:**
 (1) Remaining global scene changes; (2) Unclear fusion mechanism for conditioning; (3) Use of SD-2.1 instead of modern backbones like Flux.

**Our response:**
 – The same visualization bug identified for Reviewer Mwtn accounted for the noted global changes; we corrected figures and added new disentanglement demonstrations.
 – We clarified the fusion mechanism: all conditioning modalities are encoded and then concatenated channel-wise before being passed into the image model.
 – We agreed that stronger backbones could improve visual quality but noted that backbone choice is orthogonal to the core contribution.
 – Corresponding manuscript revisions were incorporated.

# Reviewer PGAs

**Main concerns:**
 Background changes in Fig. 1 raising questions about stability of non-edited regions.

**Our response:**
 – Again, the root cause was the visualization bug. After correction, examples show proper background consistency.
 – We added additional figures demonstrating stable disentanglement of object and camera pose across multiple scenes.
 – No further conceptual issues were raised by the reviewer.

# Reviewer sNZo

**Main concerns:**
 (1) Fig. 1 undermining disentanglement; (2) Missing ablations, especially for geometry; (3) Insufficient visual examples in the main paper; (4) Clarification about a “crop-paste-inpaint” baseline.

**Our response:**
 – As with all other reviewers, the visualization-only bug explained the perceived entanglement. We added four new figures (including iterative camera→object edits) that show clean disentanglement.
 – Regarding geometry: we clarified that its purpose is not to improve generic image quality but to expose an independent axis of controllability—enabling shape-consistent editing, geometry-preserving replacement, and cross-object geometry transfer. These capabilities are not achievable with appearance+pose alone or with trivial baselines.
 – We added more visual examples to the main text.
 – On the crop-paste-inpaint point, we explained that such a baseline cannot support 3D transformations (rotation/translation), partial modality specification, or geometry-aware operations, and is therefore not directly comparable.
 – After the reviewer noted that the revised manuscript was not visible on OpenReview, we confirmed the updates were included in the new upload.

---

**Overall**, the rebuttal clarified that the primary reviewer concerns stemmed from a late-stage visualization bug rather than model behavior. We corrected figures, added new experiments demonstrating clean disentanglement, improved manuscript clarity around modality fusion and the role of geometry, and adjusted contribution claims for precision. These updates directly address all substantive concerns raised across the four reviews.

---

### Meta-Review · Area_Chair_g33Q · 2026-01-07

**Summary:**

The reviewers recognized the conceptual potential of Neural USD as a structured, object-centric conditioning framework inspired by CG standards. However, the manuscript is undermined by concerns regarding the validity of its core claims—specifically, the disentangled control of pose, appearance, and geometry. Multiple reviewers noted that qualitative results exhibited unintended global scene changes and background artifacts during object-level editing. While the authors attributed these issues to a "visualization bug" during the rebuttal, the explanation looks insufficient; specifically, Reviewer sNZo pointed out that lighting and shading inconsistencies on static background objects suggest attribute leakage, rather than performing a clean geometric transformation. Furthermore, the lack of critical ablation studies to justify the complexity of the geometry component against simpler baselines remains a major flaw. Given the unresolved doubts, I recommend the rejection of this paper.

**Reviewer Concerns:**

Several concerns remained unresolved following the rebuttal:
- Invalid disentanglement: the "visualization bug" was unconvincing. Evidence of shading and lighting changes on static background objects indicates a fundamental failure to isolate object attributes.
- Missing ablations: the authors did not provide ablations (e.g., Pose + Appearance only) or comparisons against inpainting baselines, also leaving the geometry component unverified.
- Lack of direct baselines: the absence of side-by-side visual comparisons with Neural Assets makes it hard to assess the purported qualitative improvements.

**Reviewer Scores:**

The manuscript initially received scores of (2, 4, 6, 8). During the discussion phase, reviewer sNZo maintained a negative score of 2 after reading the rebuttal, while other reviewers did not engage. I approximate a final average score of 4.5-5.5. Based on the unaddressed concerns regarding attribute leakage and the admitted errors in core figures, I find the rebuttal insufficient to move the paper toward acceptance.

---

### Decision · Program_Chairs · 2026-01-26

Reject